# Empagliflozin Treatment Attenuates Hepatic Steatosis by Promoting White Adipose Expansion in Obese TallyHo Mice

**DOI:** 10.3390/ijms23105675

**Published:** 2022-05-18

**Authors:** Ryan Kurtz, Andrew Libby, Bryce A. Jones, Komuraiah Myakala, Xiaoxin Wang, Yichien Lee, Grace Knoer, Julia N. Lo Cascio, Michaela McCormack, Grace Nguyen, Elijah N. D. Choos, Olga Rodriguez, Avi Z. Rosenberg, Suman Ranjit, Christopher Albanese, Moshe Levi, Carolyn M. Ecelbarger, Blythe D. Shepard

**Affiliations:** 1Department of Human Science, Georgetown University Medical Center, Washington, DC 20057, USA; ryan.kurtz@georgetown.edu (R.K.); jnl43@georgetown.edu (J.N.L.C.); ghn5@georgetown.edu (G.N.); enc36@georgetown.edu (E.N.D.C.); 2Division of Endocrinology, Metabolism and Diabetes, University of Colorado Anschutz Medical Campus, Aurora, CO 80012, USA; andrew.libby@ucdenver.edu; 3Department of Pharmacology and Physiology, Georgetown University Medical Center, Washington, DC 20057, USA; baj46@georgetown.edu; 4Department of Biochemistry and Molecular & Cellular Biology, Georgetown University Medical Center, Washington, DC 20057, USA; komuraiah.myakala@georgetown.edu (K.M.); xiaoxin.wang@georgetown.edu (X.W.); suman.ranjit@georgetown.edu (S.R.); moshe.levi@georgetown.edu (M.L.); 5Department Oncology, Lombardi Comprehensive Cancer Center, Georgetown University Medical Center, Washington, DC 20057, USA; yl285@georgetown.edu (Y.L.); gak47@georgetown.edu (G.K.); michaela.mccormack00@gmail.com (M.M.); olga.rodriguez@georgetown.edu (O.R.); albanese@georgetown.edu (C.A.); 6Center for Translational Imaging, Georgetown University Medical Center, Washington, DC 20057, USA; 7Department of Pathology, Johns Hopkins School of Medicine, Baltimore, MD 21205, USA; arosen34@jh.edu; 8Microscopy & Imaging Shared Resources, Georgetown University Medical Center, Washington, DC 20057, USA; 9Department of Radiology, Georgetown University Medical Center, Washington, DC 20057, USA; 10Department of Medicine, Georgetown University Medical Center, Washington, DC 20057, USA; ecelbarc@georgetown.edu

**Keywords:** SGLT2 inhibitors, steatosis, white adipose, brown adipose, TallyHo mouse, NAFLD

## Abstract

Sodium-glucose co-transporters (SGLTs) serve to reabsorb glucose in the kidney. Recently, these transporters, mainly SGLT2, have emerged as new therapeutic targets for patients with diabetes and kidney disease; by inhibiting glucose reabsorption, they promote glycosuria, weight loss, and improve glucose tolerance. They have also been linked to cardiac protection and mitigation of liver injury. However, to date, the mechanism(s) by which SGLT2 inhibition promotes systemic improvements is not fully appreciated. Using an obese TallyHo mouse model which recapitulates the human condition of diabetes and nonalcoholic fatty liver disease (NAFLD), we sought to determine how modulation of renal glucose handling impacts liver structure and function. Apart from an attenuation of hyperglycemia, Empagliflozin was found to decrease circulating triglycerides and lipid accumulation in the liver in male TallyHo mice. This correlated with lowered hepatic cholesterol esters. Using in vivo MRI analysis, we further determined that the reduction in hepatic steatosis in male TallyHo mice was associated with an increase in nuchal white fat indicative of “healthy adipose expansion”. Notably, this whitening of the adipose came at the expense of brown adipose tissue. Collectively, these data indicate that the modulation of renal glucose handling has systemic effects and may be useful as a treatment option for NAFLD and steatohepatitis.

## 1. Introduction

According to the Centers for Disease Control, diabetes is currently the seventh leading cause of death in the United States. Moreover, this disorder has emerged as a world-wide epidemic with its prevalence only continuing to rise at an alarming rate [1]. Out of all cases, >90% of diabetics are classified as having Type 2 Diabetes (T2D) largely due to a combination of lifestyle and genetic risk factors [1,2,3]. The development of nonalcoholic fatty liver disease (NAFLD) is also on the rise and roughly half of all persons with T2D present with this disorder [4,5,6,7,8]. NAFLD is classified as a rise in hepatic steatosis that cannot be explained by alcohol consumption, immune disorders, and/or prescription of lipid-altering medications [4,5,6,7]. While its pathogenesis is incompletely understood, it is associated with insulin resistance, dysbiosis, inflammation, cholestasis, and hyperlipidemia, all of which can promote the development of nonalcoholic steatohepatitis (NASH), fibrosis, and cirrhosis, which in turn can lead to liver failure and hepatocellular carcinoma [5,6,7,9]. While not exclusively associated with T2D, NAFLD is a prevailing comorbidity.

For patients with T2D, maintaining glucose homeostasis and glycemic control is crucial to blunting the development of many diabetic conditions. Following a meal, dietary glucose is absorbed along the lumen of the small intestine in a sodium-dependent manner by the sodium glucose co-transporter 1 (SGLT1) and subsequently enters the circulation via facilitative glucose transporters (GLUTs). This glucose can then be taken up by peripheral tissues in an insulin-dependent manner. Additionally, all blood glucose is continuously filtered by the kidney. Under euglycemic conditions, all filtered glucose is recovered in the renal proximal tubule by both SGLT2 and SGLT1 [10,11,12,13].

Recently, SGLT inhibition has been used therapeutically to control blood glucose levels in persons with T2D, with SGLT2 inhibitors effectively inducing glycosuria and thereby lowering the amount of glucose that is reabsorbed. While many of the positive outcomes of SGLT inhibition can be directly linked to the kidney (attenuation of glomerular hyperfiltration and hyperglycemia), a number of its beneficial effects are extra-renal and positively contribute to cardiovascular and hepatic functions [14,15,16,17,18,19,20]. Indeed, NAFLD itself is associated with a higher risk of cardiovascular disease [21,22], making this patient population particularly well-suited to take advantage of the cardiac protection that SGLT2 inhibitors provide. Previous data from our group and others indicate that SGLT2 inhibition reduces hepatic injury, fibrosis, lipid accumulation, and inflammation in Western Diet-fed mice [19]. This correlated with decreased serum aspartate aminotransferase (AST) and alanine aminotransferase (ALT) levels and decreased expression of fatty acid synthesis master gene, sterol regulatory binding protein-1c (SREBP-1c) and carbohydrate-responsive element binding protein B (ChREBP-β). Thus far, however, this hepatic protection has been limited to descriptive studies and it is not clear if the beneficial effects of SGLT2 inhibitors are due to direct effects on the liver, or systemic changes stemming from the alteration of renal glucose handling.

To address these questions, we utilized the TallyHo mouse model that was derived from two obese outbred male mice that spontaneously developed polyuria, glycosuria, hyperinsulinemia, and hyperglycemia [23]. Similar to humans, these mice harbor several quantitative trait loci that are associated with hypercholesterolemia, diabetes, and weight gain [24]. To determine if SGLT2 inhibition promotes improved liver function in the obese TallyHo mice, we challenged both male and females, as well as two control strains—SWR/J (86% genetically identical) and C57BL6J (frequently used in diet-induced obesity studies)—with a high milk-fat diet in the continued presence or absence of Empagliflozin, one of the most well-characterized SGLT2 inhibitors on the market [25]. The high milk-fat diet was selected over the more traditional lard-based diet due to the concentration of saturated fatty acids (63%), which is known to induce greater insulin resistance, hyperglycemia, and importantly, hepatic steatosis [26,27,28]. Using these model systems, we determined that Empagliflozin reduced hepatic lipid accumulation independent of weight loss in male TallyHo mice, and increased white adipose expansion that facilitates the healthier storage of lipids and mitigates extra-adipose lipidemia. Collectively, our results indicate that by promoting glycosuria and improved hyperglycemia, SGLT2 inhibitors effectively shift the processes of fatty acid accumulation and metabolism in order to protect against liver steatosis.

## 2. Results

### 2.1. Empagliflozin Attenuates Visceral Adiposity Independent of Weight Loss

To investigate the extra-renal effects of SGLT2 inhibition, we challenged both male and female TallyHo, SWR/J, and C57BL6J mice with a high milk-fat diet (60% kcal) either alone, or with 10 mg Empagliflozin/kg body weight (based on estimated food intake) added directly into the food for 12 weeks. As expected, both male and female TallyHo mice weighed significantly more at the start of the study (*p* < 0.0001 compared to C57BL6J and SWR/J mice) and this trend continued throughout (Figure 1A). Nonetheless, Empagliflozin did not significantly alter overall body weight in any mouse strain indicating that any beneficial effects observed are independent of a change in body weight (Figure 1A). The lack of weight change may be due, at least in part, to an increase in food intake as measured in the TallyHo mice (Figure 1B). It should be noted that the female SWR/J mice did not tolerate the high milk-fat diet as well as the other strains, exhibiting greasy and unkempt fur. While none of the female mice exhibited elevated blood glucose values and were not affected by Empagliflozin treatment, SGLT2 inhibition did normalize the elevated refed blood glucose in the TallyHo males (Figure 1C). The reduced post-prandial blood glucose values correlated with a reduction in glycated hemoglobin in the TallyHo male, but not female, mice (Figure 1D).

At the conclusion of the 12-week study, MRI was performed on a subset of the mice to detect changes in abdominal adiposity (Figure 2). SGLT2 inhibition did not alter abdominal volume (Figure 2A,B) or the total visceral fat volume (Figure 2C) in any of the male mouse strains. Notably, TallyHo male mice exhibited a decreased abdominal volume compared to the control strains (Figure 2B). This aligns with the identification of a QTL allele on chromosome 4 within the gene *Tafat* that confers decreased adiposity as compared to C57BL/6J mice (Jax Labs). On the other hand, the female TallyHo mice did present with increased abdominal volume (Figure 2D,E) and visceral fat volume (Figure 2F) compared to the control strains. Nevertheless, Empagliflozin did not alter total abdominal adipose volume in any strain.

While MRI did not reveal changes in abdominal adipose volume, at the histological level, adipocyte droplets appeared larger in the TallyHo mice as compared to the C57BL6J and SWR/J controls (Figure 3A); diameter measurements of individual adipocytes revealed that Empagliflozin decreased the overall droplet size in the TallyHo male mice fed Empagliflozin (Figure 3B). This correlated with a decrease in circulating triglycerides (*p* = 0.049; Figure 3C). Crown-like structures, which are a hallmark of inflammatory processes within adipocytes, were also elevated in the TallyHo male mice compared to the control strains but Empagliflozin did not alter inflammation and macrophage recruitment (Figure 3D). No changes were noted in the visceral adipocytes of the female mice (Figure 3E,F) nor were there marked changes in circulating triglycerides (Figure 3G). Collectively, these data identified a trend where Empagliflozin increases lipolysis in TallyHo male mice, resulting in smaller individual adipocytes and a reduction in triglycerides while maintaining body weight and abdominal volume.

### 2.2. Empagliflozin Decrease Hepatic Lipid Accumulation

While Empagliflozin’s renal and cardiac protection is well-documented [25,29,30,31,32,33], its systemic effects are less appreciated. In the subset of mice that were examined by MRI (Figure 4A), we also noted a reduction in fat content in the livers of TallyHo male mice (*p* = 0.051; Figure 4B). Intriguingly, there was an increase in fat accumulation in the C57BL6J male mice (*p* = 0.036); no significant change was observed in the SWR/J mice (Figure 4B). While the female mice had overall lower degrees of detectable hepatic lipids compared to the male counterparts, Empagliflozin did not have an appreciable effect on hepatic fat levels in the TallyHo females; however, a non-significant increase (*p* = 0.052) was observed in the SWR/J females that may be due to their overall healthier appearance as compared to those who received the high milk-fat diet alone (Figure 4C).

To confirm the changes in hepatic liver triglyceride content, all livers underwent histological examination (Figure 5). In the C57BL6J and TallyHo males, all those examined developed steatosis (6/6 and 4/4, respectively), while 3/5 reached the threshold in the SWR/J mice (Figure 5A). In the female mice, 5/5 TallyHo females had severe steatosis, while only 1/3 C57BL6J females presented with lipid accumulation. No steatosis was detected in the female SWR/J mice (Figure 5B). In all cases, Empagliflozin reduced—but did not eliminate—the hepatic steatosis as observed by histological analysis (Figure 5 shows steatosis scores).

To determine if the improvements in steatosis scores by Empagliflozin correlated with changes in inflammation and liver injury, we used Second Harmonic Generation (SHG) imaging with Phasor analysis to visualize SHG generated from the collagen fibers, a marker of fibrosis (Appendix A) and assessed for changes in hepatic and circulating cytokines (Appendix A). Collagen fibers have a non-centrosymmetric structure that can generate the specific SHG signal at 370 nm emission when excited at 740 nm. Both male C57BL6J and TallyHo livers were imaged by exciting at 740 nm; while a strong signal was noted around vesicular structures in all samples, the SHG score (calculated from the ratio of the area covered by SHG to the total area of the sample, measured from fluorescence images) was low in both the control and Empagliflozin-treated mice indicating that 12 weeks of milk-fat diet feeding was not sufficient to induce fibrosis (Appendix A). This was confirmed by visualizing picrosirius red-stained livers with polarized light (Appendix A). Consistent with this finding, 12 weeks of treatment was not enough time to induce significant inflammation. Liver lysates from male and female C57BL/6J and TallyHo mice were pooled according to treatment and sex to screen for 40 different inflammatory cytokines (Appendix A). Overall, while there was some variation in cytokine expression based on genotype and treatment, Empagliflozin did not dramatically alter the profile of these cytokines. It should be noted that the overall expression of these factors was low, indicating that 12 weeks of the high milk-fat diet did not induce significant inflammation. These findings were confirmed by examination of circulating levels of CCL2 and CXCL1, two notable cytokines that mediate the recruitment of macrophages, monocytes, and T cells (CCL2) and neutrophils (CXCL1). Neither chemotactic factor was altered by Empagliflozin treatment (Appendix A).

Given the lack of fibrosis and inflammation, we also found that 12 weeks of a high milk-fat diet was not enough time to impact liver function. Gadoxetate disodium (EOVIST, Bayer) is a 726-dalton hepatobiliary MRI contrast agent that is taken up and cleared by the liver and kidneys via organic anion transporter 1 (OAT1) and multidrug resistant protein 2 (MRP2), with the dynamics of this contrast agent linked to liver function [34,35,36]. Subcutaneous injection of EOVIST and subsequent MRI revealed a strong uptake of this agent in the livers of all animals (Appendix A). When the intensity of EOVIST within the liver was analyzed over time, it appeared to rise faster in the initial 10 min window in the Empagliflozin-treated TallyHo male mice (Appendix A). This trend continued throughout the entire 30 min recording window, with the area under the curve shifting higher in the treated mice (*p* = 0.11). Despite this, when the entire curve was fitted, no significant changes in the maximum intensity, time-to-peak, or the time that was required to achieve half-maximal intensity (Km) were observed (Appendix A).

To further define the mechanisms associated with the decreased steatosis and to better describe the differential effects on different lipids, liver homogenates were assessed for triglycerides and cholesterol. Due to the minimal steatosis detected within the livers of male and female SWR/J mice, we focused our subsequent biochemical analysis on the male C57BL6J control strain and the diabetic TallyHo mice. As expected, the TallyHo mice presented with elevated hepatic triglycerides (Two-Way ANOVA strain variation *p* = 0.043; Figure 6A) and cholesterol (Two-Way ANOVA strain variation *p* = 0.015; Figure 6B) as compared to the C57BL6J mice. However, these were not changed by Empagliflozin treatment. To reconcile this discrepancy, we analyzed both free and esterized cholesterol (Figure 6C,D). In this case, Empagliflozin was responsible for a significant decrease in cholesterol esters, possibly indicating that the overall size, and not the quantity, of the droplets may be reduced (Figure 6D). Empagliflozin also decreased gene expression of several lipogenic genes: fatty acid synthase (*Fas*) in the C57BL6J mice (*p* = 0.002; Figure 6E) and *Chrebpβ* in the TallyHo mice (*p* = 0.0002; Figure 6G); no other expression changes were noted in pathways associated with lipolysis, fatty acid uptake, or β-oxidation (Figure 6H–L).

Using the neutral lipid stain Oil Red O, we noted a significant decrease in droplet size in Empagliflozin-treated C57BL6J male mice (Figure 7). However, Empagliflozin did not alter the overall area of Oil Red O nor were changes observed in the TallyHo males or in the female strains (Figure 7).

The perilipin family of lipid droplet surface proteins are required for the development of lipid droplets and several reports have suggested that hepatic perilipin-2 (adipophilin) is the most crucial player in the development of hepatic steatosis [37]. Thus, we examined expression of this critical protein (Figure 8). While there was variability in the protein expression of perilipin-2, a downward trend was noted in the Empagliflozin-treated male and female mice (both C57BL6J and TallyHo) that failed to reach statistical significance (Figure 8A,B). Similar decreases were noted in the relative expression of Plin2 mRNA in males (Figure 8C) with C57BL6J vs. TallyHo Empagliflozin-treated mice showing significant differences in expression.

### 2.3. Empagliflozin Promotes Nuchal White Adipose Expansion by Decreasing Brown Fat

While Empagliflozin did not alter total visceral adiposity, MRI analysis revealed alterations in the nuchal region. The nuchal region comprises both white and brown fat and can be detected through MRI (Figure 9A,B). Images revealed an overall whitening of the brown fat region in the TallyHo-treated mice (Figure 9A,B). To quantify this, the intensity of the nuchal white fat was calculated; no differences were noted in either the C57BL6J or TallyHo mice (Figure 9C,D). When the “brown region of interest” was measured, however, we noted an overall increase in intensity that matched the white fat (Figure 9C,D). When normalized to the intensity of the white fat region, our quantification revealed that indeed there was an overall “whitening” of the region that is typically brown fat in Empagliflozin-treated TallyHo male mice. This can also be noted in the histological examination where the adipose droplets appeared larger and less compressed in the treated mice indicative of a whitening of the brown fat (Figure 9E,F).

Uncoupling Protein 1 (UCP1) is abundant in the brown fat depots; to confirm the alteration in white and brown fat within the nuchal region, these tissue sections were subjected to UCP1 immunohistochemistry. Overall, we observed a slight decrease in the percentage of area that was stained with UCP1 but this did not reach significance in the TallyHo male mice (Figure 10A,B). In the female mice, Empagliflozin appeared to increase UCP1 area in the C57BL6J mice, indicating that its effects may be strain- and sex-dependent (Figure 10C,D).

## 3. Discussion

The protective effects of SGLT2 inhibition are now widely accepted [25,29,30,31,32,33]; however, the extra-renal and extra-cardiac effects are not well documented. Here, we show that modulation of renal glucose handling, via Empagliflozin, results in an attenuation in hepatic steatosis in TallyHo male mice fed a diet rich in saturated fat. This reduction in lipid accumulation was documented by both histological and in vivo imaging coupled with biochemical analysis. Reductions in visceral adipocyte size and a promotion of healthy, subcutaneous adipose expansion also occurred. Collectively, our data suggest that SGLT2 inhibition can limit exogenous lipid accumulation independent of changes in total body weight to promote hepatoprotection.

Obesity is a serious public health threat worldwide that results in a multitude of comorbidities including hypertension, cancer, cardiovascular disease, osteoarthritis, and liver injury. It is estimated that obesity will inflict 20% of the worldwide population by 2025 and clearly new pharmacological treatments will be required to combat this disorder [38]. Despite this prevalence, obesity is not a “one size fits all” disorder; indeed, there remains a spectrum of pathologies. Clinically, obesity is separated into monogenic obesity that is linked to clear, delineated inheritance patterns, and polygenic obesity that is multifactorial with strong influence from a variety of genetic polymorphisms and environmental factors [38]. The prevalence of monogenic obesity is quite low and yet most pre-clinical models fall under this category. This includes the well-characterized leptin-/leptin receptor-deficient mice. However, from a clinical perspective, only 63 cases of congenital leptin deficiency have ever been reported [38]. Thus, rodent models that better capture the human condition are necessary. In this study, we opted to use the TallyHo mouse. This mouse model recapitulates what is frequently observed in obese patients; they spontaneously develop polyuria, glycosuria, insulin intolerance, and hyperglycemia [23,24]. TallyHo is a polygenic mouse model where several quantitative trait loci have been linked to heightened cholesterol, diabetes, weight gain, and endothelial dysfunction. Nonetheless, fed a normal chow, the pathologies are not 100% penetrant. Thus, this study was performed using the polygenic TallyHo mice in combination with the dietary manipulation of a high milk-fat diet. Much like obesity, not all fats are created equal. Studies have revealed that the *composition* of dietary fat can greatly alter the pathophysiological state of obesity-associated diseases including cardiovascular disease, hypertension, T2D, and NAFLD [26,27,28]. Saturated fatty acids, mainly myristate and palmitate, have been linked to increased risk of steatosis, ER stress, larger liver mass, and the release of pro-inflammatory cytokines. On the other hand, unsaturated fatty acids have been shown to protect from some of these deleterious consequences. The more traditional lard-based diets that are found in high fat diets are made up of 63% unsaturated fatty acids. As a consequence, these diets do increase circulating levels of non-esterified fatty acids but do not always promote liver injury. The milk-fat diet, made up of 63% saturated fatty acids, is a more direct approach at challenging the liver [26,27,28]. This diet has been linked to increased blood glucose, elevated fasting insulin, impaired glucose tolerance, and heightened HOMA-IR. Moreover, the plasma lipidome observed upon consumption of a high milk-fat diet matches what is observed in obese individuals. Using these two tools—the TallyHo mouse in combination with a diet rich in saturated fatty acids—establishes a new model that has significant value not only for liver-related injuries, but also for other obesogenic disorders including diabetic kidney disease and cardiometabolic dysfunction.

Due to the polygenic nature of the TallyHo mouse model, there is no perfect control strain for these studies. We opted to use the SWR/J mice that are 86% genetically identical to the TallyHo mice alongside the well-characterized C57BL/6J line that is used frequently in diet-induced obesity studies [39,40,41]. Both of these strains have been used as controls for TallyHo mice in previous studies. The strains were age-matched and all had access to the same diet and drug. Nonetheless, our results indicate that Empagliflozin had the greatest impact on the TallyHo male mice. These mice had a reduction in post-prandial blood glucose, lowered circulating triglycerides and liver cholesterol esters, smaller white adipocyte diameters, and a reduction in brown fat. The improved phenotype in the Empagliflozin-treated TallyHo male mice may simply be due to the lack of diabetes development in the lean, control strains (based on blood glucose values). Even without an overt diabetic phenotype, the high milk-fat diet was sufficient to induce steatosis (to varying degrees) in all strains of mice—and Empagliflozin was able to improve this phenotype. The length of time that the mice were fed the high milk-fat diet (12 weeks) and genetic background of the mouse models both likely contributed to these differences. It should be noted that the SWR/J mice did not tolerate the diet that well. The female mice were particularly malnourished and had greasy and unkempt fur. As they age, the SWR/J mice are prone to renal deficiency brought on in part due to the development of diabetes insipidus [42]. Thus, for these studies and others that rely on renal function, the direct comparisons between the TallyHo and C57BL/6J mice may be more scientifically justified.

Finally, apart from the strain differences, we also uncovered clear sex differences in our studies. For the most part, the female mice—including the obese TallyHos—did not develop a robust diabetic phenotype and there were no detectable differences upon treatment with Empagliflozin. This aligns with the growing body of literature that points to sex differences within the development and progression of diabetes in both animal models and patient populations [43,44,45,46,47,48]. Studies have linked these differences to the protective role of estrogen and estrogen receptor signaling [46,49,50]. In light of this, more attention should be given to the role that sex plays in our interpretation of therapeutic interventions. This is especially important given the clear discrepancies between men and women when it comes to side effects of diabetic drugs [51]. In particular, adverse effects of SGLT2 inhibitors are more commonly observed in women than men. This includes a higher risk of developing genitourinary infections, ketoacidosis, and a rare occurrence of pruritus [51,52,53,54]. Despite this, however, men and women are equally protected from cardiovascular events and the drug remains safe for both genders.

White adipose depots store excess fat and can undergo a dynamic expansion and retraction to adjust for changes in whole-body metabolism. Adipocytes are known to secrete a number of factors including peptides, metabolites, and hormones in order to regulate lipolysis, lipogenesis, and fatty acid uptake [55]. These factors also contribute to insulin sensitivity and overall cardiometabolic health. Despite the characterization as “white fat”, it is clear that the individual fat depots are unique in their metabolic and gene expression profiles. Collectively, this tissue has been segmented into visceral and subcutaneous/peripheral fat based on its localization within the body. By definition, visceral fat lines organ systems and an increase in these fat pads is strongly associated with pro-inflammatory cytokines, insulin resistance, high blood pressure, dyslipidemia, and cardiometabolic dysfunction [56,57,58]. In this study, we found that Empagliflozin decreased visceral adipocyte size and circulating triglycerides, providing a link between SGLT2 inhibition and improved outcomes (Figure 2 and Figure 3). On the other hand, recent reports have suggested that subcutaneous or peripheral fat may have some protective properties [55,56,57,58]. Subcutaneous white fat has been shown to sequester excess energy under conditions of diet-induced obesity. Thus, an expansion in this fat pad may actually serve to protect tissues from adverse lipid accumulation. In support of this, we find that Empagliflozin increases nuchal white adipose tissue by decreasing nuchal brown fat depots (Figure 9). Indeed, other studies have shown that increasing the capacity within this adipose depot correlates with diminished liver steatosis, independent of glucose and insulin tolerance [59]. Taken together, our data suggest that alteration of renal glucose handling promotes healthy adipose expansion thereby reducing lipid accumulation within tissues such as the visceral fat and liver, in the male TallyHo mice. This expansion has been linked to several signaling pathways including STAT3 and growth hormone receptor [55,59,60]. Clearly, future work is needed to determine if Empagliflozin, either directly or indirectly, influences these pathways that regulate expansion and white adipose lipogenesis.

With the whitening of the nuchal region, MRI analysis revealed a decrease in nuchal brown fat in the Empagliflozin-treated TallyHo male mice (Figure 9). This was accompanied by a trending decrease in total UCP1-positive staining area. Brown fat has long been associated with non-shivering thermogenesis, which promotes heat generation, especially in cold temperatures [61]. This is mediated, in part, by the extensive network of mitochondrial cristae that are the hallmark characteristic of this tissue. On the whole, brown—and beige—adipose tissue has been associated with a protective phenotype characterized by an upregulation in antiobesity and antidiabetes pathways [61]. Given this, it is somewhat puzzling that Empagliflozin would *decrease* such a vital tissue. The measurements were performed via MRI, a well-characterized, noninvasive method for detecting changes in adipose tissue [62]. It may be that the reduction in brown fat is simply a consequence of the increased white fat area. Indeed, a recent study examining the role that growth hormone receptor plays in adiposity observed a similar phenotype to our treated TallyHo mice [60]. The authors found that loss of growth hormone receptor within adipocytes leads to an expansion of white adipose tissue which in turn decreases blood glucose values and reduces brown adipose depots. Additionally, it is known that Thyroid Hormone similarly regulates thermogenesis [63]. Clearly, an investigation into the role that growth hormone receptor and Thyroid Hormone play in SGLT2 inhibition is warranted.

Nonetheless, there are clear discrepancies between our data and previous SGLT2 inhibitor studies that have mainly shown that Empagliflozin and similar drugs lead to an increase in brown adipose [64,65,66]. Thus, we also cannot rule out that these differential effects are due to our selection of the high milk-fat diet (in lieu of a more traditional lard-based diet) or to the selection of the TallyHo mouse model. The high milk-fat diet is rich in myristate which promotes an increase in ceramide synthesis (C14 ceramide) [26,27,28,67]. Ceramides have also been shown to decrease brown adipose mass and prevent white adipose browning [68]. This points to the diet having a strong influence on the ability, or inability, to maintain brown adipose depots. Our study showed that the Empagliflozin-treated mice did *not* lose weight, and this was likely due to an increase in food consumption (Figure 1). Thus, on the whole, these mice have been consuming more myristate that in turn may have led to even more ceramide production and a corresponding decrease in brown fat. Further studies in TallyHo mice should be performed to determine if the diminished brown fat is indeed diet-dependent, and whether or not Empagliflozin alters non-shivering thermogenesis when mice are housed in thermoneutral conditions.

Empagliflozin and other SGLT2 inhibitors are well-tolerated with minimal side effects. Recently, their use has expanded to include patients suffering from kidney disease that is not associated with diabetes [69]. Our work suggests that the role of this drug may reach another target population as well—mainly those afflicted with NAFLD. Hepatic steatosis is strongly correlated with insulin resistance, dyslipidemia, and cardiovascular disease [70]. Thus, any drug that can mitigate hepatic steatosis may have far-reaching positive side effects. Of course, the true requirement for a therapeutic is that it not only prevents the development of pathologies, but that it can reverse its course. A limitation of our study was that the drug was included in the diet throughout the entire 12 weeks. Future work should be performed to determine if Empagliflozin can similarly reverse liver injury once it has been established.

While NAFLD is reversible, it can also progress to NASH, fibrosis, and cirrhosis. After 12 weeks on the high milk-fat diet, we were unable to detect significant fibrosis in any of our mouse strains (Appendix A). Thus, the analysis of Empagliflozin’s impact was limited to steatosis. Twelve weeks is the minimal amount of time needed to induce dietary pathologies and longer time points on the diet may be needed to determine if Empagliflozin similarly alters the fibrosis and inflammatory pathways known to be upregulated under conditions of NASH [26,27,28].

## 4. Materials and Methods

### 4.1. In Vivo Mouse Studies

Male and female TallyHo, SWR/J, and C57BL6J mice (12–14 weeks of age) were obtained from Jackson Labs and bred in house. All mice were maintained in a 12:12 h light–dark cycle with water provided ad libitum. The mice were maintained on a normal chow diet from weaning to 12 weeks of age and then switched to a high milk-fat diet (Research Diets D10020404; 60% kcal from fat) in the continued absence or presence of 10 mg/kg BW Empagliflozin (MedChem Express). Diet dosing was calculated by estimating daily food intake using the Research Diets custom diet food calculator. Mice were weighed weekly for the 12 weeks of the study. For the initial cohort, all three strains were used (*n* = 6/treatment/strain). At weeks 0, 6, and 12, the mice were fasted overnight (~5 p.m.–8 a.m.) and then re-fed for 2 h (~8–10 a.m.) before blood glucose measurements were obtained using a glucometer (Roche). Following the initial cohort, follow-up studies were performed only in TallyHo male and female mice (*n* = 5/treatment). These mice were used to increase the total number of mice analyzed and to calculate food intake and glycated hemoglobin. To calculate daily food intake, the food hopper was measured every 24 h for 1 week and averaged. At the conclusion of the 12-week study, blood was collected into heparin-coated tubes and used to measure glycated hemoglobin levels according to the manufacturer’s protocol (Crystal Chem). Blood was further spun at 2000× *g* for 15 min to collect plasma, that was used in the biochemical assays described below. For both cohorts, mice were euthanized (~2–4 p.m.) with an overdose of ketamine/xylazine and tissues were harvested and either flash frozen (for qPCR), drop fixed in 10% buffered formalin (for histological analysis), embedded in OCT (for Oil Red O imagining), or lysed directly into lysis buffer (for Western blotting).

### 4.2. In Vivo Magnetic Resonance Imaging (MRI)

Magnetic resonance imaging (MRI) was performed in the Georgetown-Lombardi Preclinical Imaging Research Laboratory on either a 7T/20 Avance III/ParaVision 5 or a 7T/30 USR Avance NEO/ParaVision 360 (S10 OD025153) scanner. The mice were anesthetized (1.5% isoflurane in a gas mixture of 30% oxygen and 70% nitrous oxide) and placed on a custom-manufactured (ASI Instruments, Warren, MI, USA) stereotaxic device, with built-in temperature and cardio-respiratory monitoring as described [71], and compatible with a 40 mm Bruker mouse body volume coil.

MRI of abdominal and liver fat was performed with a three-dimensional rapid acquisition with rapid enhancement (3D-RARE) sequence in the coronal orientation with the following parameters: TR: 2855 ms; TE: 12 ms; RARE Factor: 4; Matrix: 220 × 220; FOV: 50 mm × 40 mm; Averages: 4; Slice thickness: 0.75 mm; Slices: 50. Respiratory gating quantification of visceral fat depots in the imaging datasets was performed by thresholding and voxel-counting with ImageJ software (NIH), as described previously [71]. Briefly, we used a maximum intensity projection algorithm of the 3D-reconstructed image with an intensity threshold intended to segment fat only. The abdominal region analyzed was defined by superior and inferior anatomical landmarks, that is, the proximal border of the left kidney and the convergence of the left and right common iliac veins, respectively. The lateral landmark was the abdominal wall, avoiding subcutaneous fat. The percentage area corresponding to fat depots within the abdomen was calculated via the sum of the visceral fat voxels versus total abdominal voxels. Mouse liver fat was measured by quantifying the mean intensity of the region of interest (ROI) localized on the mouse liver placed in homogenous areas, avoiding structures such as large vessels and ducts. ROIs on three separate slices were selected and averaged for each mouse. For imaging of the nuchal fat, a 3D T1-weighted RARE sequence in the sagittal orientation was run with TR: 2437 ms, TE: 15 ms, FA: 74.1, Matrix: 256 × 256, FOV: 40 mm × 40 mm, Slice thickness: 0.5 mm, Averages: 4, Slices: 35. The mean intensity of nuchal brown adipose tissue (BAT) and white adipose tissue (WAT) was measured by localizing ROIs on the BAT and the WAT. ROIs on three separate slices were selected and averaged for each mouse.

### 4.3. Dynamic Contrast-Enhanced (DCE) MRI Using Gadoxetate Disodium (EOVIST^®^/Primovist^®^, BayerHealthCare AG, D-51368 Leverkusen, Germany)

DCE-MRI was performed with a T1-weighted RARE protocol in the coronal orientation, without respiratory gating, with TE: 8.2 ms, TR: 400 ms, slice thickness: 1 mm, matrix: 128 × 128, FOV: 40 mm × 40 mm, rare factor: 2, fat suppression and a duration of 25 s. Ten baseline (pre-contrast) scans were run, followed by subcutaneous parenteral administration of 0.025 mmol/kg Eovist (Bayer). Immediately after injection, 60 MRI scans were acquired repetitively over approximately 25 min. Relative liver enhancement (RLE) was used to quantify hepatic function. Briefly, the mean intensity of ROI localized in homogenous areas of the liver were quantified in baseline and post-injection images. Liver function was calculated using the formula *RLE* = (*SI_Liver enh_* − *SI_Liver unenh_*)/*SI_Liver unenh_* × 100. The time-to-peak intensity was calculated by identifying the time at which EOVIST uptake reached its peak intensity within each ROI. Maximum intensity and Km were calculated for each individual curve by analyzing for Michaelis–Menten kinetics using GraphPad Prism. The raw curves were used for all calculations and a 2nd order smoothing with 10 neighbors was applied for visualizing (Appendix A).

### 4.4. Histology Analysis

Tissue samples (liver and adipose) were drop-fixed in 10% buffered formalin for 24 h, embedded in paraffin, and processed for Periodic Acid Schiff (PAS; liver), Picrosirius Red (PSR; liver), or Hematoxylin and Eosin (H&E; adipose) staining by the Histopathology Shared Resources at Georgetown University Lombardi Comprehensive Cancer Center. For steatosis scoring, PAS-stained livers were imaged and blindly analyzed and scored by a trained histologist (AZR). To detect fibrosis, PSR-stained livers were imaged using polarized light as described previously [72]. To analyze white adipose, the entire H&E-stained slides were scanned using an Aperio imager at 40× and visualized with ImageScope (Leica Biosystems). To measure droplet size, the diameters of >50 adipocytes/section were measured and averaged. To quantify crown-like structures (CLS), scanned sections were analyzed in OuPath software. Briefly, entire scanned sections were divided into 900 × 650 μM grid (field) system using the software in order to keep track of counted fields. Individual CLS were counted in each field. Total CLS in the section were then normalized to the number of fields quantified.

### 4.5. Second Harmonic Generation

Second harmonic generation (SHG) in tissue and biological systems is specifically sensitive to fibrillary collagen and myelin [73,74,75,76]. SHG and autofluorescence images were obtained using a modified Olympus FVMPERS (Waltham, MA, USA) microscope equipped with a Spectra-Physics Insight X3 (Milpitas, CA, USA) laser and ISS FastFLIM (Champaign, IL, USA) acquisition card. The samples were excited using two-photon excitation at 740 nm with a 20X air objective (LUCPLFLN 0.45NA, Olympus, Tokyo, Japan). The fluorescence was collected using the DIVER (Deep Imaging Via Enhanced Recovery) detector assembly and recorded using a FastFLIM card (ISS, Champaign, IL, USA) [77,78,79]. The pixel dwell time for the acquisitions was 20 µs and the images were taken with sizes of 256 × 256 pixels with a field of view of 318.8 µm (Zoom = 2X). To have high signal to noise ratio, 16 frames were collected for each area. Images were acquired for an area of 1.9 mm × 1.9 mm (6 × 6 images). The data from each pixel were collected using the passive mode, where the raster scanning was performed using the Olympus software; the images were collected using the FLIMBox/FastFLIM system and the scanning parameters were matched to ensure proper image acquisition, and analyzed using the SimFCS software (Laboratory for Fluorescence Dynamics, University of California, Irvine, CA, USA).

Liver sections (5 µm thick) were imaged using the homebuilt DIVER (Deep Imaging via Enhanced Recovery). The details of the DIVER detector have been described elsewhere [75,78,80]. Briefly, this microscope uses a forward detection scheme which is ideally suited for harmonic generation imaging [81] due to the forward direction of the harmonic signal. In this work, a combination of UG11 and BG39 filters, which creates a window of observation around 350 nm, was used for separating SHG (second harmonic generation) signal [81]. A different filter was used to separate the blue wavelength (400–500 nm) and used autofluorescence imaging. The phasor plot is calibrated using Rhodamine 110 in water which has a mono-exponential lifetime of 4.0 ns. A modified version of our earlier work [81] was used to calculate fibrosis score. SHG signal was identified by its lifetime and phasor signature as it has a lifetime of zero and it occupies s = 0, g = 1 coordinates of the phasor plot. The area covered by the SHG signal was calculated for each area of the sample image and then summed to calculate the total number of pixels covered by SHG. A fluorescence image of the same area was used to calculate the total number of pixels in the FLIM image. A ratio of these two quantities was used as a measure of extent of fibrosis. This ratiometric calculation ensures that the changes in the tissue architecture and empty areas of an image are accounted for and gives a better measure of the extent of fibrosis compared to just the calculation of amount of SHG signal.

### 4.6. Powdered Liver Samples

At the conclusion of the study, powdered liver samples were obtained in order to ensure a homogenous sample for downstream assays. To prepare the livers, one whole lobe of flash frozen liver was immersed in liquid nitrogen in a ceramic mortar and pestle. Samples were ground into a coarse powder, more liquid nitrogen was added to the mortar, and the samples were subsequently ground to a fine powder. Powdered liver was stored at −80 °C until further use (triglyceride assay, cholesterol assay, and qPCR).

### 4.7. Cholesterol Assay

Total liver cholesterol and cholesteryl ester quantification were measured from the powered liver samples using Abcam’s Cholesterol/Cholesteryl Ester Quantitation Assay kit (ab65359) following the manufacturer’s Fluorometric protocol. Briefly, 10 mg of powdered, flash frozen liver tissue was mixed with 400 μL of Chloroform: Isopropanol: NP-40 (7:11:0.1) in order to extract lipids. Following the initial spin, 200 μL of supernatant was aliquoted to a fresh Eppendorf tube and re-spun. These samples were dried at 50 °C for 1 h. The samples were then reconstituted in 400 μL of Assay Buffer.

### 4.8. Triglyceride Assay

To determine liver triglycerides, the lipid extractions that were obtained for the cholesterol assay were diluted 1:3 in Assay Buffer (ab65359). Then, 1.5 μL of standard (ThermoFisher 23-666-422), diluted tissue lysate, and a blank (buffer) were loaded into a 96-well plate, in triplicate. Next, 150 μL Infinity Triglycerides Liquid Stable Reagent (ThermoFisher TR-22421) was added to each well, and the plate was incubated for 15 min at room temperature. Following a brief, gentle shaking, absorbance was read at 540 nm using a plate reader (BMG Labtech, Ortenberg, Germany). To calculate plasma triglycerides, blood was collected in a heparinized tube and spun at 2000× *g* for 15 min to collect the plasma. Then, 1.5 μL of plasma was added per well, combined with 150 μL triglycerides stable reagent, incubated for 15 min, and absorbance was read at 540 nm as described above. All samples were loaded in triplicate.

### 4.9. Western Blotting

Fresh or flash frozen liver samples (~50 mg) were homogenized on ice in 500 µL of the following lysis buffer: 200 mL of 250 mM sucrose + 300 µL triethanolamine, with cOmplete Mini (SigmaAldrich-11836153001) and PMSF protease inhibitors added just before. When probing for phosphorylated proteins, the HALT phosphatase inhibitor cocktail was also added (ThermoFisher 78430). All samples were centrifuged at 2800× *g* for 15 min at 4 °C. Protein concentrations were established via the Bradford assay. Then, 30 µg of total protein was loaded per sample into hand-cast gels (percentages varying by target size, 7–15%). Protein expression of interest was quantified relative to ponceau stain. The Perilipin 2 primary antibody (Progen GP40) was used at 1:1000 in 5% BSA. HRP-conjugated secondary antibody (Invitrogen A18769) was incubated at 1:10,000 for 1 h at room temperature and bands were visualized with Supersignal West Pico Plus Chemiluminescent Substrate (ThermoFisher 34580) on a BioRad imager. Quantification was performed via ImageJ analysis software. Briefly, images were converted to 8-bit, and mean pixel intensity for desired bands was measured, and normalized against background pixel density.

### 4.10. Inflammatory Cytokines

At the conclusion of the study, liver lysates that were prepared for Western blotting were also used to assess the expression of 40 different cytokines using a cytokine array card (RayBio C1 array AAM-INF-1-8). Briefly, equal amounts of protein from each treatment group (control and Empagliflozin-treated TallyHo and C57BL6J male and females) were pooled and 200 μg of lysate was added to the cytokine array membrane that was pre-spotted with antibodies specific for the cytokines of interest. The assay was run according to manufacturer’s protocol and the membranes were imaged using a BioRad Imager. Quantification was performed via ImageJ analysis software where the chemiluminescent images were converted to 8-bit, and mean pixel intensity for each spot was measured and normalized against background pixel density. The density of the spot was then normalized to the positive control that was run alongside the cytokines on the same membrane. The relative expression of each cytokine was plotted and converted to a heat map using GraphPad Prism Software. To determine circulating levels of CCL2 and CXCL1, blood from TallyHo male and female mice was collected in heparin-coated tubes and spun to collect the plasma. Then, 50 μL of each sample was run in duplicate according to the manufacturer’s protocols (R&D Systems). Following development, the microplate was read at 570 nm with wavelength correction. The standard curve was plotted using a four-parameter logistic curve fit and the concentration of circulating cytokines was determined from the standard curve and expressed as pg/mL.

### 4.11. Quantitative PCR (qPCR)

RNA was isolated from powered liver samples (50 mg) using standard phenol-chloroform extraction. Then, 1 ug of cDNA was synthesized from isolated RNA using the iScript Select cDNA synthesis kit (BIO RAD 170-8896) per the manufacturer’s protocol. Quantitative PCR was performed in triplicate utilizing 2x SensiFAST SYBR No-ROX mix (Meridian-C755J00) on the Bio Molecular Systems micPCR unit (40 cycles: denature—95 °C for 5 s; anneal—65 °C or 60 °C for 5 s; extend —72 °C for 5 s). Melt curve analysis was performed following each reaction to ensure that only a single peak was detectable. Endpoint PCR was also performed to confirm that the primers amplified a single product, and the identity was confirmed via sequencing. Expression levels are presented as relative fluorescent units (RFUs) as normalized to 18s rRNA. A list of primers used in the study is included in Table 1.

### 4.12. Oil Red O

2.5 g Oil Red O power (ThermoFisher A12989) was dissolved in 400 mL of 100% isopropanol for at least 2 h at room temperature. On the day of the experiment, this solution was diluted at a 1.5:1 ratio with ddH2O and passed through a 45 μm filter. Filtered Oil Red O solution was then chilled to 4 °C and allowed to thicken. Fresh frozen 10 μm liver sections were removed from −80 °C, equilibrated to room temperature, and a hydrophobic barrier was drawn. Staining was performed as described previously [82]. Briefly, filtered, thickened Oil Red O solution was applied to coat each section. After 2 min, slides were washed 10x in gently rocking ddH_2_O. Staining was visualized using an EVOS FL Auto microscope and quantification was performed using ImageJ. For this, the images were converted to 8-bit with the threshold auto-set according to ImageJ settings. The mean pixel density as a percent area was calculated.

### 4.13. Immunohistochemistry for UCP1

Fixed, paraffin-embedded inguinal brown adipose tissue sections were deparaffinized by two 10 min incubations in Xylenes. The slides were then rehydrated in decreasing ethanol concentrations (95%, 90%, 80%, 50%, 35%) and fully submerged in TBS. Antigen retrieval was performed by submerging slides in a retrieval buffer (5 mL 0.1 M Citric Acid, 45 mL 0.1 M Na Citrate, 450 mL ddH_2_O) and microwaving (1100 W microwave) for 5 min at 70% power 3 times. Following antigen retrieval, slides were washed twice by gentle rocking for 5 min in wash buffer (1X TBS + 0.025% Tx-100). Sections were then blocked with 10% Normal Donkey Serum in 1% BSA made with 1X TBS, for 2 h at room temperature. Slides were then incubated with UCP1 (Abcam–ab10983) primary antibody in 1% BSA or simply 1% BSA for no-primary controls, at 1:100 concentration, in 4 °C overnight. The following day, slides were washed in gently rocking wash buffer, twice for 5 min at room temperature. Sections were treated with 0.3% H_2_O_2_ for 15 min. HRP-conjugated secondary (Invitrogen–A16023) diluted to 1:400 in 1% BSA was incubated on the slides for 1 h at room temperature. To develop the stain, the DAB chromogen kit (BD-550880) was used per manufacturer’s instructions. DAB reaction was subsequently quenched with ddH_2_0, and slides were mounted. Full slide scans were performed by the Histopathology & Tissue Shared Resource at Georgetown University using an Aperio slide scanner. Percent area was determined using the Aperio ImageScope software.

### 4.14. Statistics

Values presented are means +/− standard error. Two-way ANOVA followed by Sidak’s post hoc test was performed to determine differences by strain and treatment. Significance markings in the figures indicate a statistical difference as determined by the multiple comparison tests. Data were plotted and analyzed using GraphPad Prism.

## Figures and Tables

**Figure 1 ijms-23-05675-f001:**
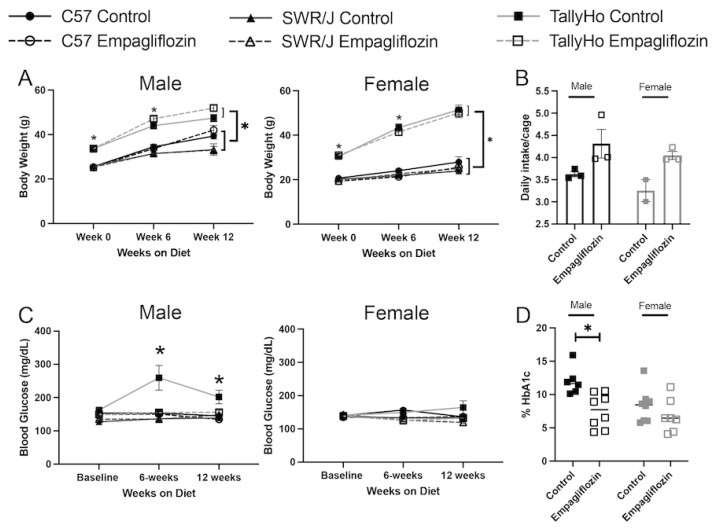
**Empagliflozin attenuates blood glucose in male TallyHo mice independent of changes in body weight.** Male and female TallyHo (squares), SWR/J (triangles), and C57BL6J (C57; circles) were fed a high milk-fat diet in the absence (filled symbols) or presence (open symbols) of 10 mg/kg BW Empagliflozin. (**A**) Body weight was tracked weekly and plotted at weeks 0, 6, and 12. * indicates *p* < 0.05 for TallyHo vs. C57BL6J and SWR/J mice. No statistical differences were detected between control and Empagliflozin-treated mice. (**B**) Daily food intake (averaged/cage) was measured for control and Empagliflozin-treated TallyHo male and female mice. (**C**) To track blood glucose values, mice were fasted overnight before being refed for 2 h. Following the refeeding, blood glucose values were measured using a glucometer. * indicates *p* < 0.05 for TallyHo controls vs. all other groups. (**D**) Glycated hemoglobin was measured for control and Empagliflozin-treated TallyHo male and female mice. * *p* < 0.05 TallyHo control vs. Empagliflozin.

**Figure 2 ijms-23-05675-f002:**
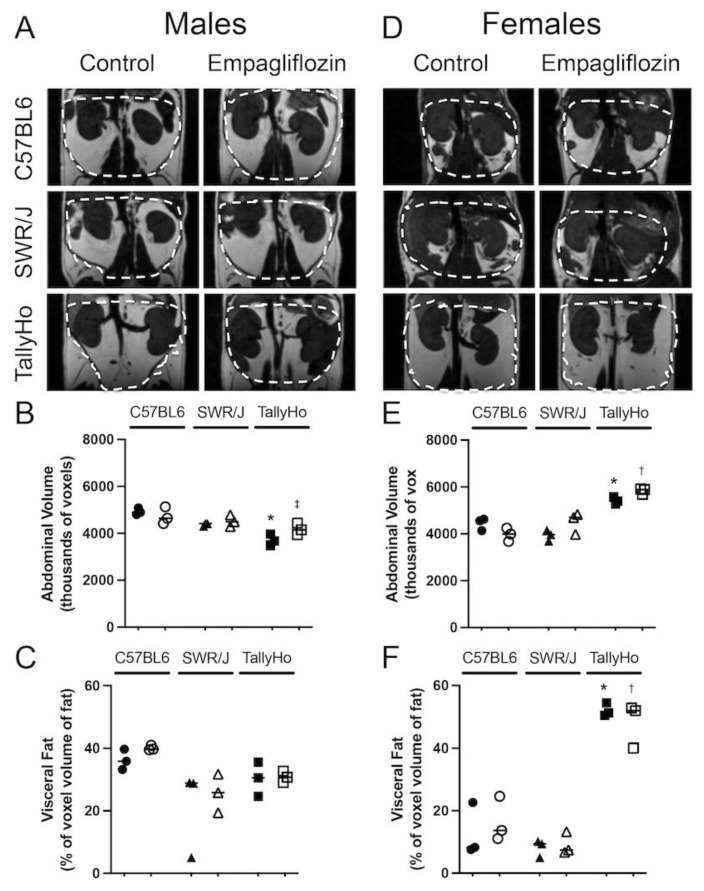
**Empagliflozin does not change visceral fat volume.** Male and Female TallyHo (squares), SWR/J (triangles), and C57BL6J (circles) were fed a high milk-fat diet in the absence (filled symbols) or presence (open symbols) of 10 mg/kg BW Empagliflozin. MRI was performed to detect changes in visceral adiposity. Representative images of the abdominal region from male (**A**) and female (**D**) mice highlight the area quantified (dashed lines). (**B**,**E**) The total volume of the abdomen was quantified and graphed as thousands of voxels. (**C**,**F**) Within the abdominal region, the total volume of visceral fat was calculated. Two-way ANOVA with multiple comparisons was performed. * *p* < 0.05 TallyHo control mice vs. C57BL6J and SWR/J; ^†^ *p* < 0.05 TallyHo Empagliflozin vs. C57BL6J and SWR/J; ^‡^ *p* < 0.05 TallyHo Empagliflozin vs. C57BL6J.

**Figure 3 ijms-23-05675-f003:**
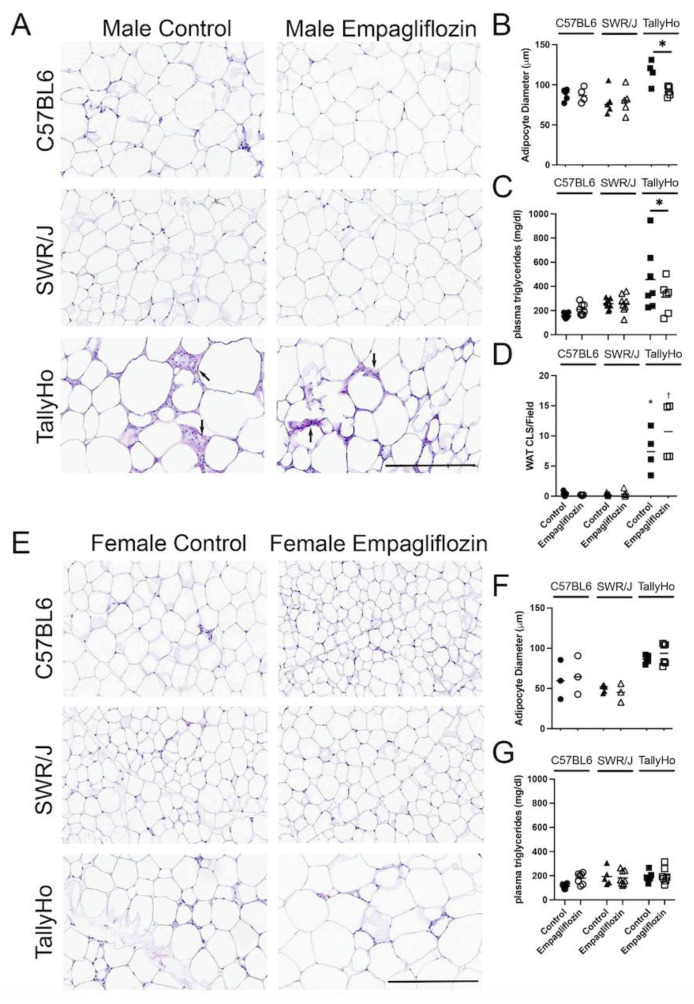
**Empagliflozin reduces white adipocyte size and circulating triglycerides in male TallyHo mice.** Male and Female TallyHo (squares), SWR/J (triangles), and C57BL6J (circles) were fed a high milk-fat diet in the absence (filled symbols) or presence (open symbols) of 10 mg/kg BW Empagliflozin. The nuchal white fat (WAT) was then isolated and subjected to H&E histological analysis (**A**,**E**). (**B**,**F**) Quantification of adipocyte diameter, (**C**,**G**) circulating triglycerides in male (**B**,**C**) and female (**F**,**G**) mice. (**D**) Quantification of macrophage infiltration through the presence of crown-like structures (CLS)/field of view (arrows note CLS). In (**B**,**C**), Empagliflozin decreased the adipocyte diameter and circulating triglycerides with * *p* < *p* = 0.05. In (**D**), both TallyHo control and Empagliflozin-treated male mice had significantly more CLS compared to the control strains (* *p* < 0.05 TallyHo control mice vs. C57BL6J and SWR/J; ^†^ *p* < 0.05 TallyHo Empagliflozin vs. C57BL6J and SWR/J). Scale bar = 50 μm.

**Figure 4 ijms-23-05675-f004:**
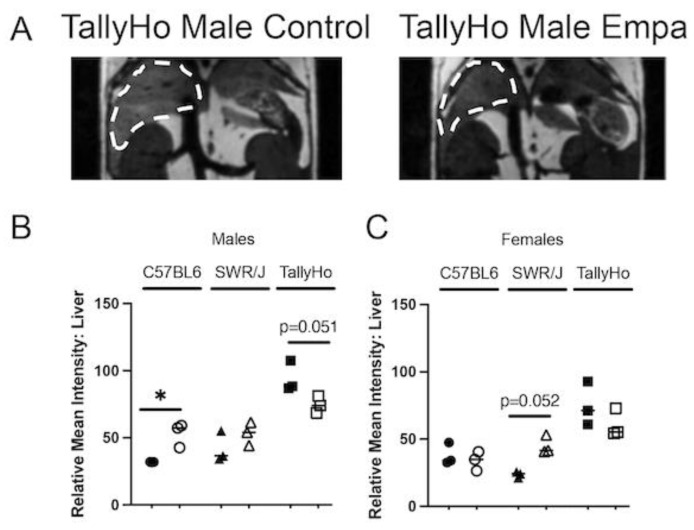
**Empagliflozin decreases liver steatosis.** MRI was performed on both male and female TallyHo, SWR/J, and C57BL6J mice following 12 weeks of Empagliflozin treatment. (**A**) Regions of lighter intensity seen in the livers is an indication of lipid accumulation (dashed outline). A representative image is shown for both a TallyHo control and Empagliflozin-treated male mouse. (**B**,**C**) The relative mean intensity of the liver region was measured for each animal ((**B**)—male; (**C**)—female). * *p* < 0.05 between control and Empagliflozin.

**Figure 5 ijms-23-05675-f005:**
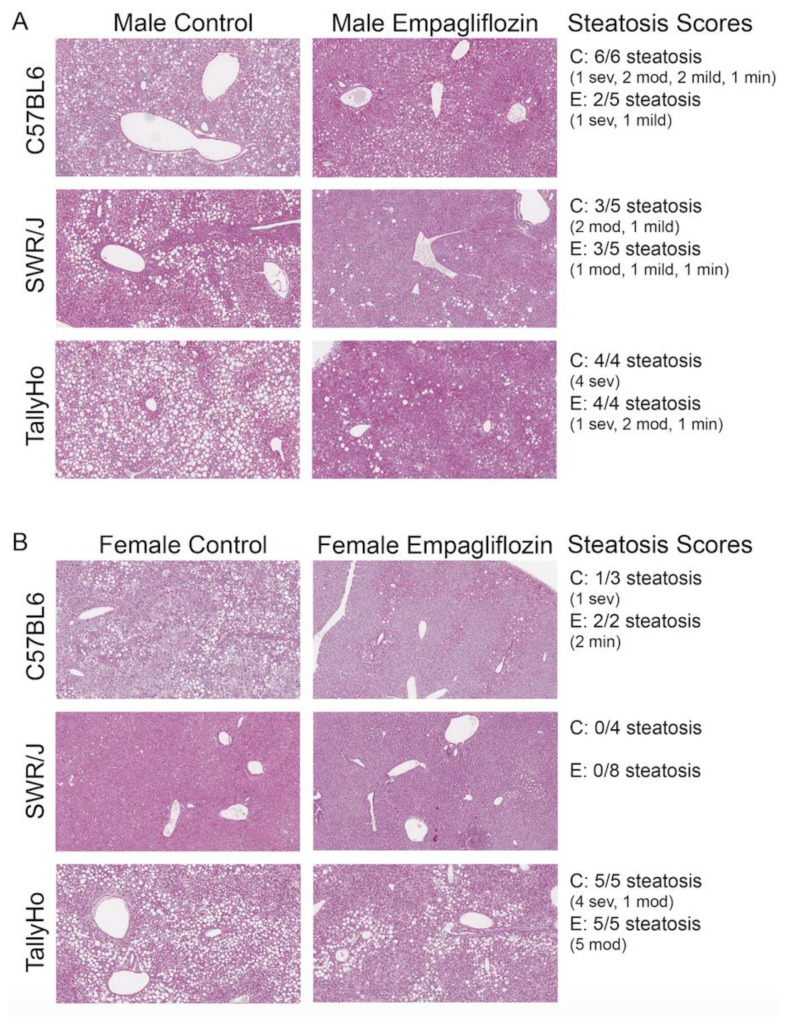
**Empagliflozin attenuates hepatic steatosis.** Livers from control and Empagliflozin-treated mice were isolated, stained for PAS, and imaged. Both male (**A**) and female (**B**) control mice fed a high milk-fat diet exhibited severe steatosis. Steatosis scores decreased in all of the Empagliflozin-treated mice with the exception of the SWR/J females where steatosis was not detected. Sev: severe; mod: moderate; min: minimal.

**Figure 6 ijms-23-05675-f006:**
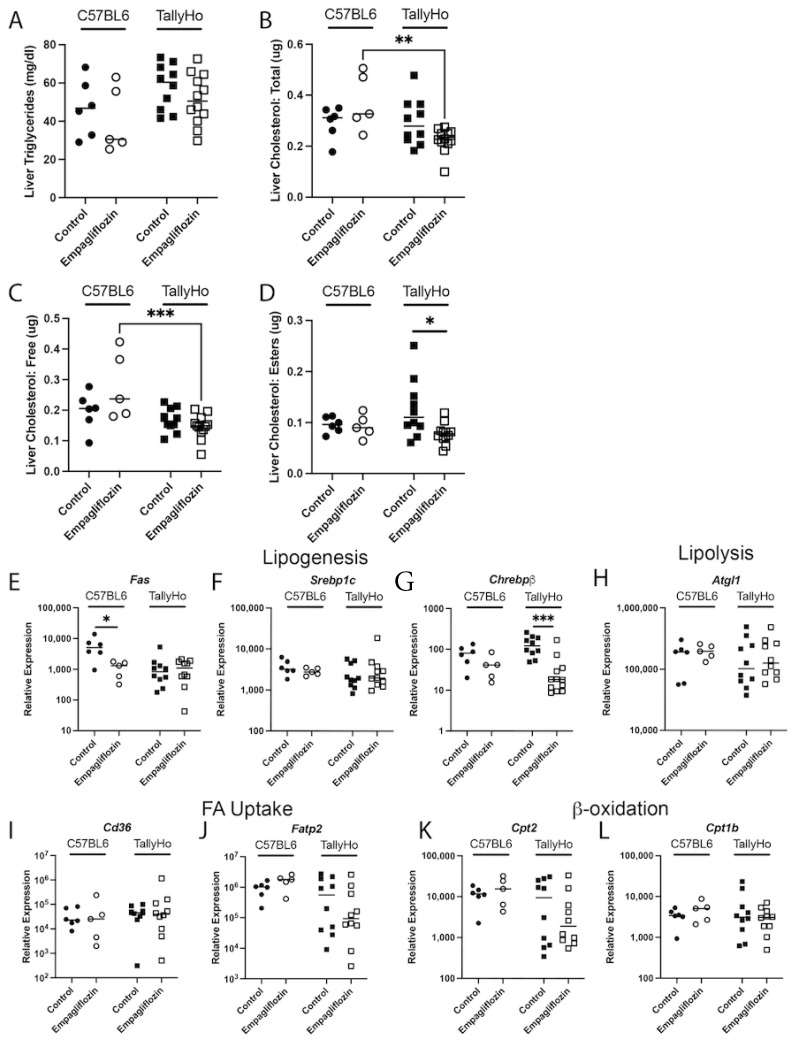
**Empagliflozin attenuates hepatic cholesterol esters in male TallyHo mice.** Livers from control and Empagliflozin-treated C57BL6J and TallyHo mice were isolated, lysed, and analyzed for total triglycerides (**A**), total cholesterol (**B**), free cholesterol (**C**), and cholesterol esters (**D**). Additionally, RNA was isolated and reverse transcribed to cDNA to perform qPCR for enzymes involved in lipogenesis (**E**–**G**), lipolysis (**H**), fatty acid (FA) uptake (**I**,**J**), and β-oxidation (**K**,**L**). * *p* < 0.05 between control and Empagliflozin-treated mice. ** *p* < 0.005 and *** *p* < 0.0005 between C57BL6J and TallyHo mice.

**Figure 7 ijms-23-05675-f007:**
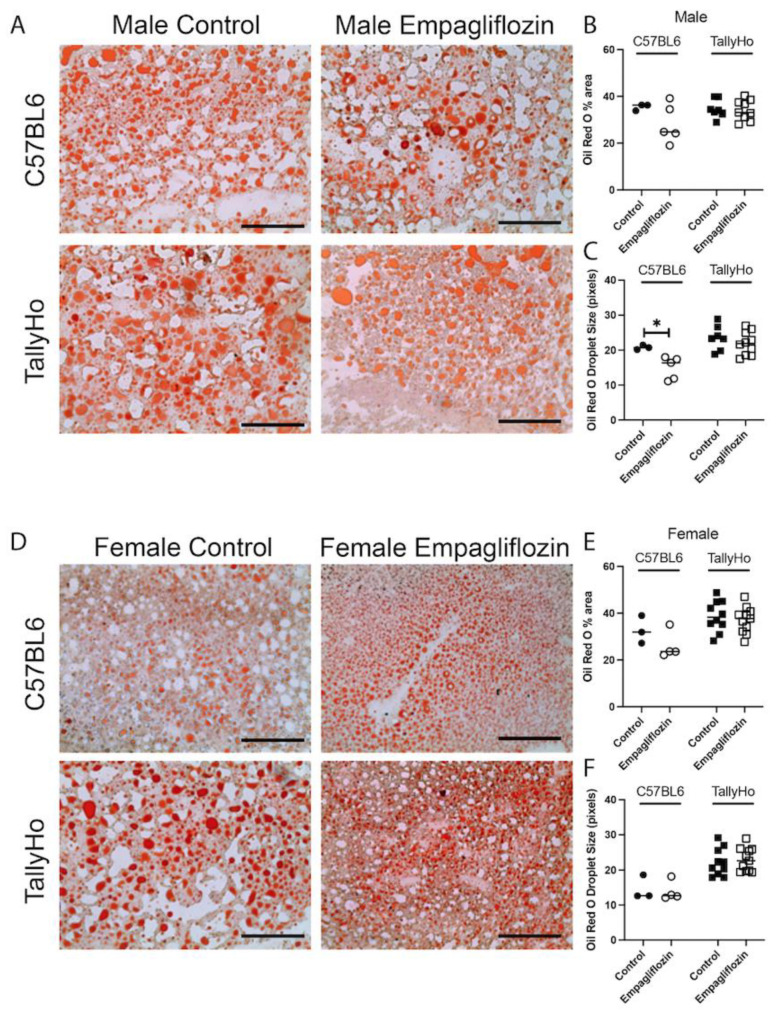
**Hepatic neutral lipids are not changed upon Empagliflozin treatment.** Livers from control and Empagliflozin-treated C57BL6J and TallyHo mice were isolated, embedded in OCT, and sectioned. Fresh frozen sections were then stained with neutral lipid stain, Oil Red O. Robust red droplet stain was noted in both the male (**A**–**C**) and female (**D**–**F**) livers. Empagliflozin did not alter the total Oil Red O area (**B**,**E**) and only decreased droplet size (**C**,**F**) in C57BL6J male mice. * *p* < 0.05. Scale bar = 200 μm.

**Figure 8 ijms-23-05675-f008:**
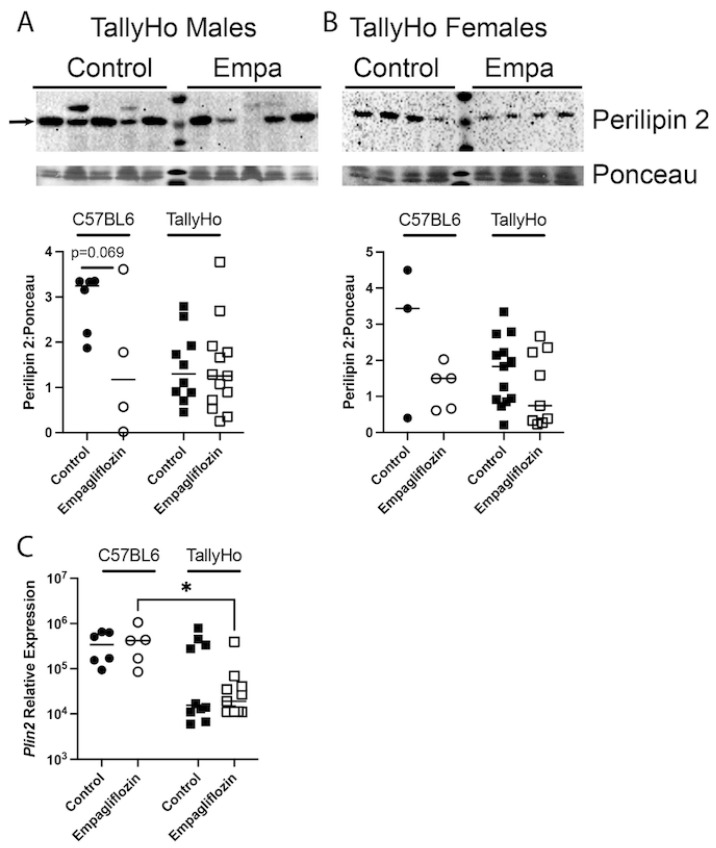
**Hepatic perilipin 2 expression is not changed upon Empagliflozin treatment.** Perilipin 2 protein expression was measured via immunoblotting in both C57BL6J and TallyHo male (**A**) and female (**B**) livers. Representative blots are shown from TallyHo livers in (**A**,**B**). Expression was normalized to total protein stain via Ponceau imaging and graphed. (**C**) qPCR for *Plin2*, the gene that encodes for Perilipin-2, in control and Empagliflozin-treated male mice. * *p* < 0.05 between C57BL6J and TallyHo mice.

**Figure 9 ijms-23-05675-f009:**
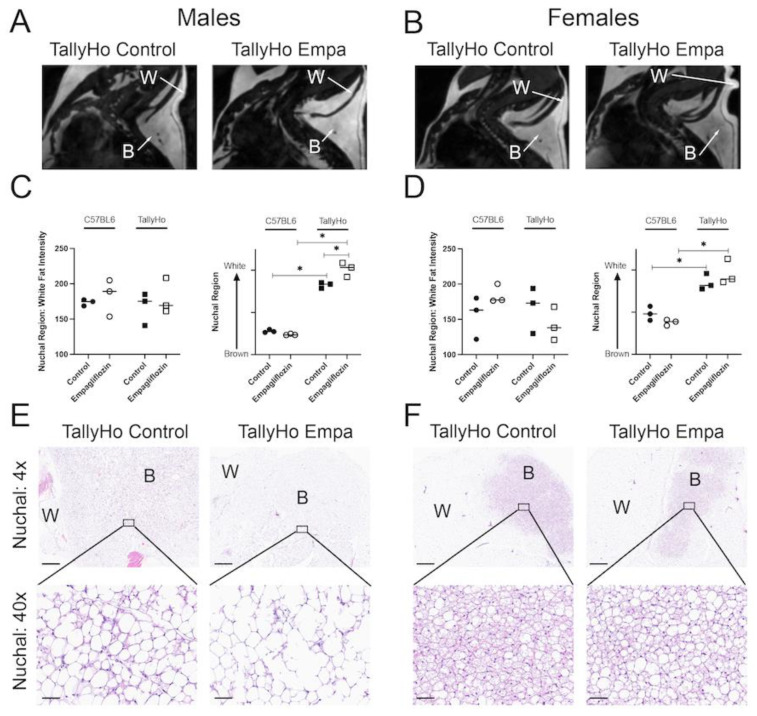
**Empagliflozin decreases nuchal brown fat in TallyHo male mice.** MRI was performed on the nuchal region of both C57BL6J and TallyHo male (**A**) and female (**B**) mice in the sagittal orientation. Representative images of this region are shown highlighting the brown adipose ROI (labeled ‘B’) and the white adipose ROI (labeled ‘W’). Mean intensity was measured for the white adipose ROI (**C**,**D**). To quantify the brown adipose ROI, the intensity of the brown region was normalized to the white region. An increase in intensity of the brown ROI is an indication that the nuchal region has increased its white adipose ((**C**,**D**) **left** graphs). (**E**,**F**) Representative H&E-stained nuchal regions are shown for the TallyHo mice at both 4x and 40x. The ‘B’ and ‘W’ regions of interest are noted. White adipose was found to be increased in Empagliflozin-treated TallyHo male mice. * *p* < 0.05 between the labeled groups calculated by 2-way ANOVA followed by multiple comparisons. Scale bar = 60 μm.

**Figure 10 ijms-23-05675-f010:**
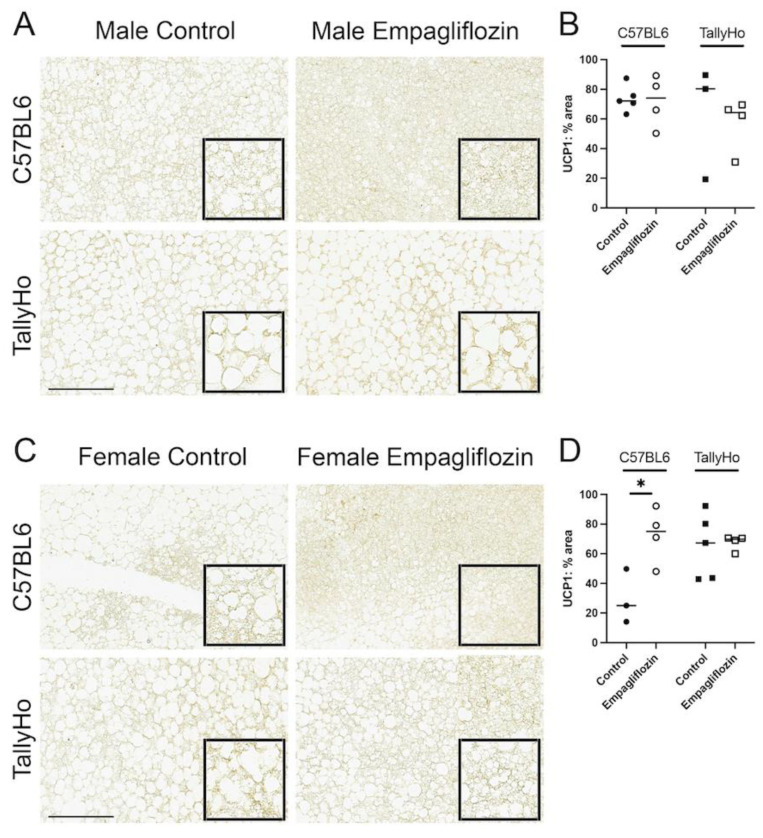
**Uncoupling protein 1 (UCP1) is slightly reduced in Empagliflozin-treated TallyHo male mice.** Brown adipose was isolated from the nuchal region, paraffin-embedded, and stained for uncoupling protein 1 (UCP1), a protein specifically enriched in brown fat in both male (**A**) and female (**C**) mice. Quantification of UCP1 staining revealed a downward trend in the total area of UCP1 (**B**,**D**) in TallyHo mice. This expression was increased in C57BL6J female mice (**D**). * *p* < 0.05; scale bar = 50 μm.

**Table 1 ijms-23-05675-t001:** Primers used in the study.

Gene Name	Forward Primer	Reverse Primer
SREBP-1C	ACCACGGAGCCATGGATTG	GGGAAGTCACTGTCTTGGTTG
Plin2	GACAGGATGGAGGAAAGACTGC	GGTAGTCGTCACCACATCCTTC
FAS	TCGTCTATACCACTGCTTACTAC	ACACCACCTGAACCTGAG
ATGL1	GCTGTGGAATGAGGACATAGGA	GCATAGTGAGTGGCTGGTGAA
CD36	TTGAAAAGTCTCGGACATTGAG	TCAGATCCGAACACAGCGTA
CPT1b	AAGAGACCCCGTAGCCATCAT	GACCCAAAACAGTATCCCAATCA
Fatp2	GATGCCGTGTCCGTCTTTTAC	GACTTCAGACCTCCACGACTC
CPT2	CAAAAGACTCATCCGCTTTGTTC	CATCACGACTGGGTTTGGGTA
ChREBPβ	TCTGCAGATCGCGTGGAG	CTTGTCCCGGCATAGCAAC
18s rRNA	GTAACCCGTTGAACCCCATT	CCATCCAATCGGTAGTAGCG

## Data Availability

Not applicable.

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
