# Peer review of "Empagliflozin Treatment Attenuates Hepatic Steatosis by Promoting White Adipose Expansion in Obese TallyHo Mice"

_ijms, 2022, doi:10.3390/ijms23105675_

Round 1

Reviewer 1 Report

The article is very interesting and reports several data. In order to improve the article, I would suggest:

  1. Please include levels of serum glycemia in different animal groups.
  2. Does empaglyfozin also affect liver inflammation? Histological and RT-PCR data investigating macrophage infiltration and cytokine expression can be included.
  3. Does empaglyfozin affect animal serum inflammation? Assessment of circulating chemokine levels (CXCL1, CCL2) has to be included.

Reviewer 2 Report

The manuscript presented by Ryan Kurtz et al. entitled “Empagliflozin treatment attenuates hepatic steatosis by promoting white adipose expansion in obese TallyHo mice” is well written, clear, and easy to read. The topic is new and of interest and therefore, it adds information on the Sodium-glucose co-transporters (SGLTs) inhibitors in the nonalcoholic fatty liver disease (NAFLD) compared with other published articles until now.

It is of great interest the result achieved in which Empagliflozin is able to reduce hepatic lipid accumulation independent of weight loss in TallyHo mice.

Minor

In line 283 please write perilipin, not in capital letter PERILIPIN

Do not add sub-sections to the discussion. This section should be harmonized with respect to your results and the literature.       
Since, SGLT2 inhibitors are on the market this section add also the possible side effects of the drugs, in relation to gender, since either your results point out different drug responses between males and females. 

Please see these references

2174/1574886311666160405110515

10.2147/DMSO.S184437

1185/03007995.2014.890925

After this changing for me can be published 

Round 2

Reviewer 1 Report

No further comments. It can be accepted.